# Contrasting effects of information sharing on common-pool resource extraction behavior: Experimental findings

**Dimitri Dubois**[1]\*, **Stefano Farolfi**[2], **Phu Nguyen-Van**[3,4], **Juliette Rouchier**[5]

**1** CEE-M, Univ. Montpellier, CNRS, INRAE, Institut Agro, Montpellier, France, **2** CIRAD UMR G-EAU, Univ. Montpellier, Montpellier, France, **3** BETA, CNRS, Univ. Strasbourg, Strasbourg, France, **4** TIMAS, Thang Long University, Hanoi, Vietnam, **5** LAMSADE, CNRS, PSL-Univ. Dauphine, Paris, France

\* dimitri.dubois@umontpellier.fr

## Abstract

This paper experimentally investigates the impact of different information sharing mechanisms in a common-pool resource game, with a view to finding a mechanism that is both efficient and inexpensive for the managing agency. More precisely, we compare the observed extraction levels produced as a result of three mechanisms: a mandatory information sharing mechanism and two voluntary information sharing mechanisms that differ in the degree of freedom given to the players. Our main result is that a voluntary information sharing mechanism could help in reaching a lower average extraction level than that observed with the mandatory mechanism.

**Data Availability Statement:** Data are available via protocols.io (https://dx.doi.org/10.17504/protocols.io.bgx6jxre).

## Introduction

In economics, goods are usually classified according to two dimensions: excludability and rivalry. A good is excludable if a person can be excluded from its consumption and is rival if its consumption by one person reduces its consumption by another. These two dimensions make it possible to classify goods into four categories: private goods (excludable and rival), club goods (excludable, but not rival), common goods (non-excludable, but rival) and public goods (non-excludable and non-rival). Public goods and common goods have the particularity of placing individual and collective interests in apparent opposition as well as creating tension in the choice of action, which is commonly referred to as a social dilemma. Indeed, rivalry in extractive common goods implies that agents may think that they should consume as much of the good as possible, fearing that the others leave nothing. Non-rivalry in public goods implies that agents have an incentive to benefit from the goods without contributing to their production. The phenomenon of over-exploitation of common goods was highlighted by Hardin in 1968 [1] with his famous "tragedy of the commons". However, it has been shown that in some real-life settings, complex property rights and institutions could enable the emergence of trust and make coordination possible, so as to attain self-governance of common goods and sustainability of the resource [2–4]. On the other hand, laboratory and field investigations on common-pool resources have shown that social information, i.e. information available to

**Funding:** This research benefited from CNRS financial support (PEPS-DIPP project: Decision, Indicators and Public Policy) attributed to J. Rouchier.

**Competing interests:** The authors have declared that no competing interests exist.

participants about the actions of the others, is a key element for managing the resource. This article contributes to the existing literature on this topic.

In the context of a public good game, [5] conducted a laboratory experiment in which they tested three treatments that differed in the information available to players. In a baseline treatment, players had no information about the actions of other members of their group at neither the aggregate nor individual level. In a second treatment, the players had information about the total contribution made by the group to the public good. Finally, in a third treatment, the players were provided with information about both the total contribution of the group and (anonymous) individual contributions. The data revealed significantly higher levels of contribution in the last treatment. The authors explained that; in a situation of complete information, it is not in the interest of individuals to forgo participation in the production of the public good. The reason for this, is that even if it had just a small influence over the contributions made by the other members of the group, opportunistic behavior would inevitably lead to lower production of the good. Signaling cooperative behavior becomes strategic in this case. A few years later [6] found nevertheless that explicit signaling through announcements did not improve the average level of contribution in the game. A reputation mechanism, i.e. an individual history of past behavior in the game visible by everyone in the group, did not either. In the context of a common-pool resource game; which is the standard representation of a common good [7, 8] tested two treatments in the laboratory. In the first treatment individuals had aggregated information (total group extraction), and in the second treatment they had the additional information on the individual choices and payoffs of their group members. The authors observed that in the complete information treatment individuals extracted more and thus moved away from the socially optimal solution. The main reason was that when complete information was available, the individuals had a tendency to imitate the best performance (in addition to imitating the average level of extraction in the group), which in this kind of game means the least cooperative individual. These studies clearly show that social information plays a very important role in social dilemmas (see also [9]). The fact that individual decisions are public in the group, even if anonymous, is interpreted by some as the possibility of sending a signal of cooperation. This can be beneficial to the group as it increases the production of the public good or improves the management of the common-pool resource. However, this information can also have harmful effects when it encourages the imitation of less cooperative choices in the group. Information about the actions taken by all the players in the group is therefore not necessarily beneficial. In this article we set out to establish a system of voluntary sharing of information that keeps the beneficial effects of the dissemination of information (signal) and mitigates its negative effects (imitation of the most selfish). Our hypothesis is that a voluntary sharing (or disclosure) information mechanism allows the creation of selective social information, or upward social information [10–12], which favors cooperation between individuals.

We set up a laboratory experiment using a common-pool resource game to measure the extent to which a voluntary sharing mechanism makes it possible to mitigate over-exploitation of the resource compared to a system in which the collection and dissemination of information, at an individual level, is automatic and mandatory. The fact that individual extractions are automatically and mandatorily made public may have a negative effect on the motivations of the individual, such as crowding-out effects [13] or boomerang effects [14] for example. Furthermore, this implies collection and dissemination of information that escapes the individual, who may then feel watched and not free to make their own choices [15]. Letting the individual choose to make their level of extraction public is likely to make them feel responsible and give them the impression that they can act on the collective norm, by voluntarily expressing what they consider to be appropriate behavior [16]. They may also feel satisfaction

in showing their cooperation in the social dilemma; the warm-glow effect [17], and acting in a prosocial manner that benefits other people [18]. If the most cooperative individuals voluntarily share their extraction decision with the other members of their group, the social information provided to the group is likely to favor the emergence of a cooperative descriptive norm [19]; as observed by [10], as well as triggering conditional cooperative behavior [20]. We therefore believe that a voluntary, rather than mandatory, information sharing mechanism leads to a higher level of cooperation within a group. In a common-pool resource game, this means lower levels of resource extraction. In addition, if there is a cost for collecting and disseminating social information, it may be useful to find a mechanism that is both more efficient and less costly [21].

Nowadays the exchange of information is extensively facilitated by connected objects and social networks. Smartphones are constantly asking us if we want to share our information and activities (sports activities, photos, videos, news, etc.). It is simple to share information within a group or community, often at the click of a button. The use of common-pool resources has also been transformed by new techniques that allow users to share and disclose real-time information about their consumption. In the electricity sector for instance, many countries have deployed smart grids to place data flow and information management at the heart of improving supply efficiency. In the case of domestic water use; particularly in Western countries, remote metering enables water services and providers to improve the efficiency of water supplies. More real-time information about consumption is also provided to water users, who can then adjust their habits and detect possible leaks. In France, 50% of domestic water users should have a remote meter connected within 3 to 4 years [22]. These systems do not yet provide water consumers with information on the consumption of others, but initiatives to establish social standards are underway. Benchmarks do already exist based on aggregations of macro-data (city, region, and nation). In California, the law on sustainable groundwater management; adopted by the State in 2004, obliges groundwater sustainability agencies (GSA) to draw up groundwater sustainability plans, as well as requiring users located outside the GSA's management area to report their extraction levels to the State Water Board. Conversely, in countries like Tunisia, Algeria and Morocco, private water wells are proliferating rapidly [23] and information on the extraction of water from these wells is hidden by irrigators to avoid paying fees or penalties. Community management of aquifers has been recognized as one of the possible instruments for groundwater management [24], but it can only be achieved if all users commit to a cooperative scheme and share information on their private consumption. Even so, both management costs and the attitudes of the users depend on the policy implemented. Several voluntary information disclosure initiatives have started to develop, such as the Carbon Disclosure Program in the field of forest degradation. This program requires its members to disclose how they address forest risk commodities in their supply chain [25]. A voluntary information sharing mechanism can be an important tool for regulators, as it does not require that the collection of private information be organized by adding technical tools, therefore making it much less expensive. Up until now though, its effectiveness is yet to be proven. The purpose of this article is to test, in the laboratory, the effects of two voluntary information sharing mechanisms in the context of decentralized management of a common resource. Testing the mechanism in the laboratory is a first step [26, 27], a field experiment will be necessary in a second step to test its external validity [28].

We experimentally test two treatments based on voluntary information sharing and compare the observations to a reference treatment in which information is collected and disclosed in an automatic and mandatory manner. In the first treatment, after making their extraction decision, the players have to decide whether or not they wish to make this extraction decision public. In the second treatment, players have to decide whether or not they wish to make their

extraction level public, as well as additionally deciding the amount of extraction they wish to publicize. [21] experimentally tested two voluntary sharing mechanisms in a public good game. Her first treatment was identical to ours, where after choosing their level of contribution to the public good, the players had to decide whether or not they wished their contribution to be made public to the group. In her second treatment, players decided whether or not to publicize their contribution decision before making it. [21] showed that these two treatments led to higher levels of contribution than a treatment in which contributions were automatically and mandatorily made public. However, she did not observe any difference between the two voluntary sharing systems.

The contribution of our article is threefold. Firstly, we contribute to the literature on the role and impact of social information in social dilemmas, more specifically in the context of a common-pool resource. Secondly, we confirm that the results of [21] can be transposed into the context of the common-pool resource game. Thirdly, we shed light on the effects of a new voluntary sharing mechanism, which gives more freedom to individuals. Our first result is confirmation that the mechanism of voluntary sharing of information on the chosen level of extraction reduces the phenomenon of over-exploitation of the resource compared to automatic and mandatory sharing. Our second result is that a voluntary information sharing mechanism in which an individual decides the value of the information disclosed is just as effective as a mechanism that does not allow this freedom. It does however introduce strategic behaviors on the part of the least cooperative individuals, which in the long run, may lessen the positive effect of voluntary sharing.

## Materials and methods

### Experimental design

In a Common Pool Resource game (hereafter CPR), each player $i$ in a group of $N$ players can extract from $y_i = 0$ to $y_i = E$ tokens from a common resource that contains $NxE$ tokens. $E$ was equal to 10 in our experiment. For each extracted token, player $i$ earned 3 ECUs (Experimental Currency Unit), but created a negative externality for each of the other group members. In our experimental game, the payoff function of player $i$ was given by $\pi_i(y_i, Y) = 3y_i - 0.01875Y^2$ where $Y = \Sigma_i\, y_i$ and $y_i$ is the individual amount extracted by player $i$.

To avoid a corner solution (zero-unit extraction as the socially optimal solution), we adapted an existing model [29] by transforming the linear payoff function into a quadratic one. Our game had features that ensured a social dilemma where individual and collective interests were divergent. These features were: (i) whatever the amount extracted by the other group members, player $i$ had a higher payoff when extracting the maximum (10), and whatever the amount extracted by $i$, the payoff was higher when the other group members extracted nothing from the common resource, with the dominant strategy therefore being to extract the maximum possible (10); (ii) the collective payoff, computed as the sum of individual payoffs, was maximized when the total amount extracted by the group was 20 tokens, with a symmetric issue where each player extracted exactly 5 tokens.

We tested two treatments with voluntary information sharing mechanisms in comparison to a reference treatment with automatic and mandatory information sharing. More precisely, in this baseline treatment; called MD for *Mandatory Disclosure*, the player's extraction was automatically and mandatorily displayed on the summary screen of each member of their group. In the two voluntary sharing treatments, the player's extraction was displayed on the summary screen of each other member of their group only if they had previously chosen to share such information. The difference between the two voluntary sharing treatments was that in one treatment; called FD for *Free Disclosure*, the player could also decide the value to be

**Table 1. Summary view of the treatments.**

|  | MD | VD | FD |
|---|---|---|---|
| Voluntary sharing | No | Yes | Yes |
| Freedom to choose the value to be disclosed | No | No | Yes |

displayed on the screens of the other members of their group while in the other treatment; called VD for *Voluntary Disclosure*, the player did not have this option. Therefore, in VD if the player had decided to make their decision public, the value seen by the other group members was the player's actual extraction. In FD, this was not necessarily the case, as it could have been the actual extraction or a different value altogether, since the player was free to decide what value to display. Table 1 summarizes the treatments.

The game used in all three treatments was the one described in the first paragraph of this section, with fixed groups of four players formed randomly at the beginning of the game. The game consisted of 20 rounds, with each round divided into three stages in the two voluntary sharing treatments (extraction decision, disclosure decision and summary) and two stages in the MD treatment (extraction decision and summary). In the extraction decision stage, the player had to decide how many tokens to extract from the collective account, an integer between 0 and 10. In the disclosure decision stage (only in treatments VD and FD, skipped in treatment MD because the player didn't have this choice) the player had to decide whether or not they wanted their extraction to be displayed on the summary screen of their group members, with the additional option in FD treatment to enter the value to be displayed. In the summary stage, the screen displayed (for the current round): the player's extraction level; the total amount extracted by their group and their payoff for the round. In the MD treatment the players were also informed of the individual extractions of all members of their group. In the VD treatment the players were informed of the individual extractions of the group members who had chosen to disclose this information. Likewise for FD treatments, but here the players knew that the displayed values had been entered by the group members themselves. From each screen the players had access to the history of the previous rounds. The history screen included the information corresponding to each past round. Explicitly (for each past round): the extraction of the player; the total extraction of the group; the individual extractions of the members of the group (all or some of them); the payoff of the round and the cumulative payoff since the first round. Two important elements need to be specified. Firstly that in the disclosure decision stage, when deciding whether to make their extraction decision public or not (and its value in the FD treatment), the players had information on the total quantity extracted by their group in the extraction stage. The second specification being that on the summary screen, regardless of the treatment, the individual extractions were anonymous and this was common knowledge. Indeed, on the one hand there was no associated identifier and furthermore it was specified that these values were displayed in a random order each round. Table 2 gives a summary

**Table 2. Stages of one round, in the three treatments.**

|  | MD | VD | FD |
|---|---|---|---|
| *Extraction decision stage* | Player decided how much they extracted from the collective account | | |
| *Disclosure decision stage* | Skipped, the player had no choice | Player decided whether their extraction would be displayed on the round summary screen of their group members | Player decided whether their extraction would be displayed on the round summary screen of their group members, and if so, entered the value that would be displayed |
| *Summary stage* | Round summary | | |

view of the different steps depending on the treatment. The instructions of the experiment are available at https://dx.doi.org/10.17504/protocols.io.bgxzjxp6.

The experiment took place at the Montpellier Laboratory for Experimental Economics (LEEM) in France. The experimental protocol was presented to the LEEM working group, which ensured that the protocol was in compliance with the rules of experimental economics and the corresponding ethical rules. The latter gave its agreement for the experiment to be carried out within the LEEM platform. We organized six sessions (two per treatment), each with 16 or 20 participants. A total of 104 subjects took part in the experiment. The mean age of the participants was 26 years (std. 6.58, median age 24 years), with 56.73% of women and 43.27% of men. The participants were students from various disciplines of the university of Montpellier randomly selected from a pool of nearly 3,000 volunteers handled with the Online Recruitment Software for Economic Experiments (ORSEE, [30]). We made sure that none of the participants had previously participated in a common-pool resource game. The experiment took place on a computer. Each subject was in an individual box with partitions around them to ensure anonymity of decisions. The sessions lasted approximately one and a half hours, including initial instructions and payments. The average payment in the game was € 13.

## Conjectures

The first conjecture states that even if there is no direct reward associated with sharing, many players voluntarily share their extraction decision. By no direct reward associated with sharing, we mean that there was no monetary gain in the game associated with the action of sharing its extraction. The payoffs of the game depended solely on the extraction choices of the player and the members of their group.

**Conjecture 1: Individuals voluntarily disclose extractions with significant frequency.**
There are several possible motivations behind a player's choice to disclose their decision, as explained by [21]. These motivations include the willingness to signal an intention to cooperate [5]; a warm-glow effect [17]; a feeling about the right level of extraction in the game, i.e. the extraction that is socially appropriate [16], or the descriptive norm that should apply in the group [19]. In the Voluntary Disclosure treatment, since the values disclosed are the actual extractions, non-cooperative individuals have no interest in making their extractions public if they think they might negatively influence the choice of others. In addition, some studies have shown that individuals who do not behave in a pro-social manner may feel guilty or ashamed when it is observable by others ([31–33], see also [34] for a theoretical model). Therefore, we expect that the voluntary sharing mechanism of the VD treatment will favor the emergence of upward social information i.e. cooperative individuals disclose their extractions and non-cooperative individuals keep them private.

To corroborate this, we should observe in the VD treatment that the disclosed extractions are lower compared to the non-disclosed ones (Conjecture 2.1). Given the arguments underpinning conjectures 1 and 2.1, there are several reasons to believe that extractions in the VD treatment will be lower compared to extractions in the MD treatment (Conjecture 2.2). First, we know from several studies that almost 50% of individuals are conditional cooperators [20, 35–37]. If these conditional cooperators observe cooperative decisions, they are likely to make cooperative decisions as well. Second, it has been shown that social information favors the convergence of decisions towards observed decisions [10–12, 38, 39], and that this imitation dynamic lies behind the formation of social norms [19, 40].

Conjecture 2: In the Voluntary Disclosure treatment:

1. *the disclosed extractions are lower than the non-disclosed ones*

2. *the average extractions are lower than in the Mandatory Disclosure treatment*

The effect of the *Free Disclosure* mechanism isn't so straightforward. On one hand, cooperative players are supposed to consent to make their decision public and to disclose the amount actually extracted, while non-cooperative players are supposed to prefer to hide their extraction (as in the VD mechanism). As a result, the disclosed extractions should be lower than the non-disclosed ones (Conjecture 3.1). Yet on the other hand, some of the cooperative players may use the freedom to choose the value disclosed to signal the socially optimum solution, even if their actual extraction is a little higher. This is an attempt not to lose too much money while the (symmetric) socially optimal solution is not achieved. In addition, non-cooperative players can hide behind a false declared extraction; lower than the actual one, to benefit from the possible influence of this signal on the other members of the group (Conjecture 3.2). [41] shows that for some people their concern over appearing honest and cooperative may outweigh their desire to actually be honest and cooperative. These people prefer lying to appearing selfish. This is common knowledge, which can call into question the self-selection mechanism induced by voluntary sharing mentioned in the argumentation of conjecture 2.1. If this strategic behavior is detected, it can: undermine trust in the group; prevent the formation of a group norm (or norm compliance) and cause conditional cooperators to reduce their cooperation. Overall, the *Free Disclosure* mechanism is likely to create noise in the disclosed social information and thus cause difficulties for the group in managing the common-pool resource. As it stands, we are not able to predict the outcome of the *Free Disclosure* treatment with certainty, as it partly depends on the composition of the group (the number of unconditional cooperators, conditional cooperators, and non-cooperators). We nevertheless expect that the average extractions would be lower than in the *Mandatory Disclosure* treatment (Conjecture 3.3), because on average only one third of people are non-cooperative (free-riders, [20]), and probably not all are willing to lie to gain more on the backs of others.

Conjecture 3: In the Free Disclosure treatment:

1. *the disclosed extractions are lower than the non-disclosed extractions*

2. *the disclosed extractions are lower than the actual extractions*

3. *the average extractions are lower than in the Mandatory Disclosure treatment*

## Results

When voluntary disclosure was offered to players (VD and FD treatments), most of them decided to make their decision public; as shown in Fig 1, in support of our first conjecture. More specifically, in the VD treatment, more than 30% of players chose to disclose their extraction decision 100% of the time. This percentage dropped to 16% in the FD treatment but; as can be seen in the figure, between 5 and 15% of players revealed their decision at least 10 periods out of 20 (50% of the time). The graph at the bottom of the figure shows that the frequency of information sharing was relatively constant over time (between 60% and 85%) with a decrease however in the VD treatment that was not observed in the FD treatment.

### Treatment effect

Table 3 provides statistics on the average level of extraction depending on the treatment and Fig 2 displays the average extraction trends for the three treatments. From the first round, without any prior information about the other members' behavior in the group, the VD treatment stood out from the MD and FD treatments with a lower average extraction level (7.31 vs 8.06 and 8.03 respectively). This initial effect persisted all throughout the game, even though

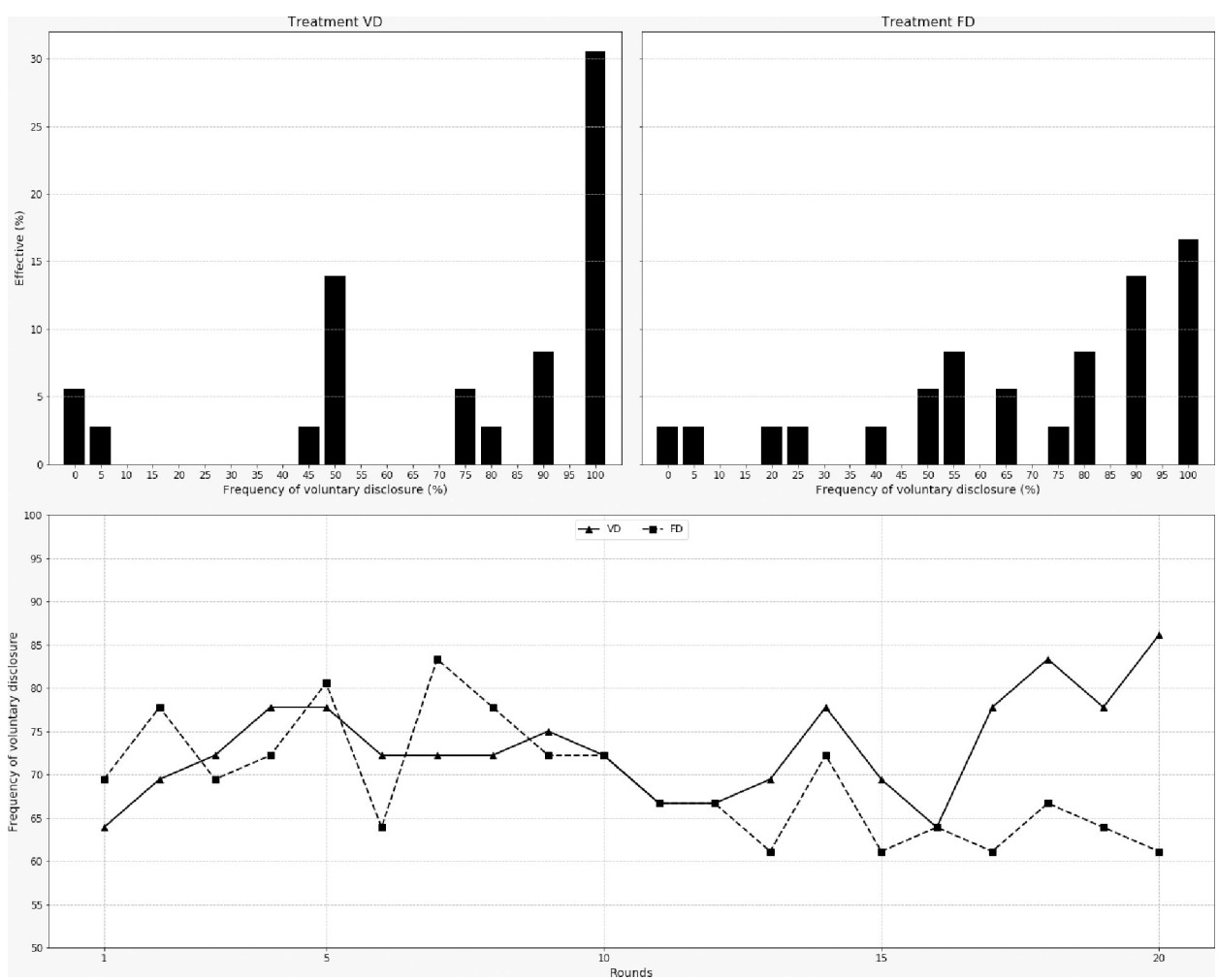

**Fig 1. Frequency of voluntary disclosure.**

with replications the three treatments converged towards the choice of the dominant strategy, as is often the case in experimental social dilemma games [42, 43]. We observe in Fig 2 that the average extraction in the MD treatment was almost always higher than in the other two treatments. Non-parametric tests (Wilcoxon rank-sum test) comparing the average extractions of the three treatments confirmed that the averages in the MD treatment were significantly higher than in the VD treatment (rank-sum test statistic = 4.193, p-value < .001) and also higher than in the FD treatment (rank-sum test statistic = 3.111, p-value = .002), supporting conjectures 2.2 and 3.3. On the other hand, the two voluntary sharing treatments were not significantly different from each other (rank-sum test statistic = -1.258, p-value = .208).

**Table 3. Summary statistics.**

| Treatment | # Groups | Average extraction (std) | | | |
|---|---|---|---|---|---|
| | | Round 1 | Rounds 1 to 10 | Rounds 11 to 20 | Rounds 1 to 20 |
| MD | 8 | 8.06 (2.20) | 9.00 (1.83) | 9.45 (1.46) | 9.22 (1.67) |
| VD | 9 | 7.31 (2.66) | 7.84 (2.55) | 8.86 (2.31) | 8.35 (2.49) |
| FD | 9 | 8.03 (2.43) | 8.26 (2.40) | 9.06 (1.77) | 8.66 (2.14) |

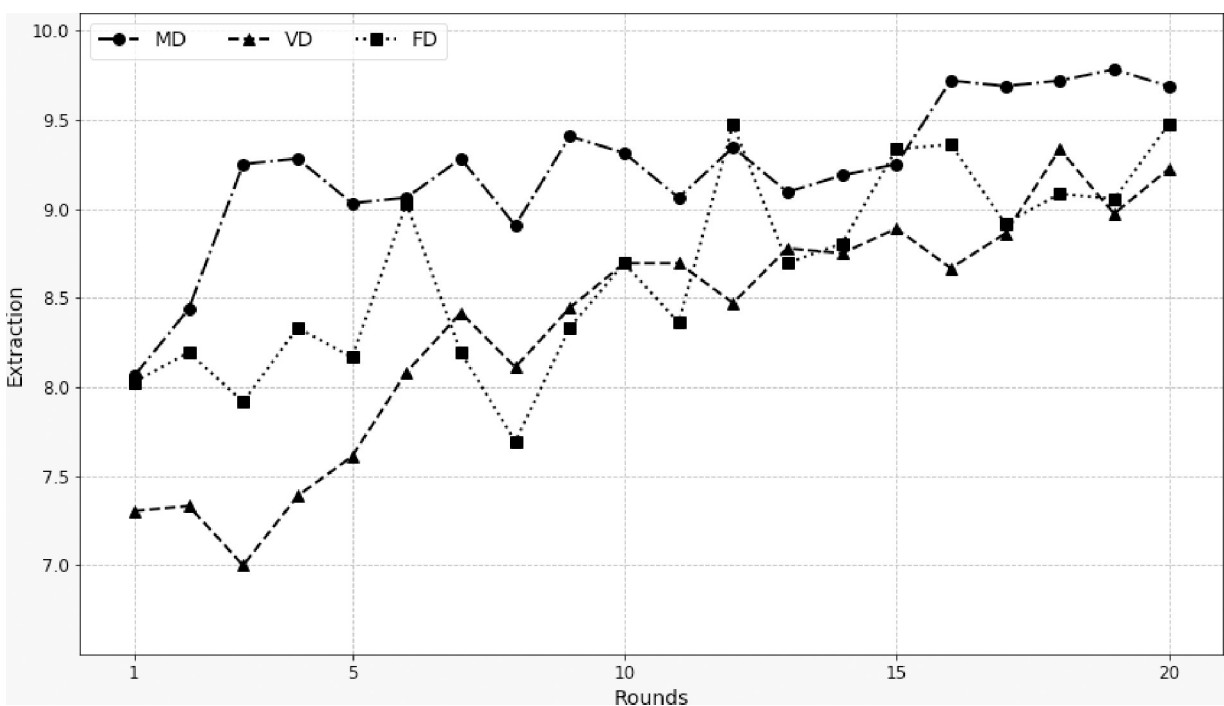

**Fig 2. Evolution of average extraction per treatment.**

To further analyze the data, we consider the following econometric model. Let $y_{it}$ be the extraction of player $i$ in round $t$. This amount is both left and right-censored, i.e. $0 \leq y_{it} \leq 10$. We have a dynamic panel data model: $y_{it} = \rho y_{i,t-1} + x_{it}'b + \mu_i + \varepsilon_{it}$; $i = 1, 2, \ldots, N$; $t = 1, 2, \ldots, T$, where $y_{i,t-1}$ is the extraction of player $i$ in the previous round, $x_{it}$ corresponds to the whole set of explanatory variables including both time-variant variables (total extraction of player $i$'s group in $t-1$; decision-making time; information-related variables such as dummy of information sharing and the number of individuals who disclosed their decision in the previous round) and time-invariant variables (treatment dummy variables). The main concern of our model is that during the experiment players can learn about the decision process. We think that past individual and group decisions can have an impact on the individual's current decision. This thus corresponds to the concept of state dependence, persistence in individual decisions over time, or learning effects often highlighted in the literature. It is recognized that the dynamic nature of the model is related to the well-known initial conditions problem leading to the inconsistency of traditional estimators in panel data econometrics (see for example [44, 45]). Note that the regression error term is composed of two parts: an idiosyncratic error $\varepsilon_{it}$ and an individual-specific effect $\mu_i$. Following [45], the initial conditions problem can be fixed by specifying a more general model where the $\mu_i$ are defined as correlated random effects with the following assumption: $\mu_i \mid y_{i1}, z_i \sim N(\alpha_0 + \alpha_1 y_{i1} + z_i'\gamma, \sigma_\mu^2)$. This assumption appears to be general enough, as it suggests that an individual-specific effect depends not only on the initial extracted amount $y_{i1}$, but also on a set of values of explanatory variables ($z_i \equiv x_{i1}, \ldots, x_{iT}$). The model with this assumption therefore corresponds to a dynamic Tobit model for panel data with correlated random effects (CRE). It results that the whole set of explanatory variables of our model includes: the lagged individual extraction $y_{i,t-1}$; the control variables included in $x_{it}$; the initial individual decision $y_{i1}$ and the set of auxiliary regressors $z_i$.

A comparison between nonlinear dynamic models with correlated random effects and it's fixed effects counterpart was beyond the scope of this study (a reference to [45, 46], among others will provide more details on this issue). However, we can list at least three advantages of the CRE approach. Firstly, it specifies a more general distribution for the individual effects that can be correlated with the regressors. Secondly, this approach makes it possible to calculate the average partial (or marginal) effects. Finally, the endogeneity of some regressors can be conveniently handled by using the control function approach of [46]. Furthermore, estimation of the CRE dynamic Tobit model, compared to the dynamic Tobit model with standard random effects, implies two additional sets of variables: initial decision ($y_{i1}$) and a set of auxiliary variables ($z_i$). A likelihood-ratio (LR) test was performed to compare the two models. The null hypothesis corresponded to $\alpha_1 = \gamma = 0$. For the whole sample (all treatments included), the test statistic was 275.93 and the p-value of the chi-squared distribution with 58 degrees of freedom was close to 0, leading to the rejection of the original model in favor of the dynamic Tobit model with correlated random effects. This test shows the importance of the initial observation problem, which has to be controlled for. The significance (at the 10% level) of this coefficient ($\alpha_1$, Table 4) provides an illustration of this result, which reveals the presence of an anchoring

**Table 4.  Estimation results for the whole sample using the CRE dynamic Tobit model with individual extraction as the dependent variable.**

| Variable | Coefficient | Partial effect |
|---|---|---|
|  | (Std.Err.) | (Std.Err.) |
| Individual past decision | 0.006 | 0.002 |
|  | (0.061) | (0.021) |
| Group past decision | 0.133** | 0.046** |
|  | (0.030) | (0.010) |
| Decision-making time | -0.049** | -0.017** |
|  | (0.010) | (0.003) |
| Treatment VD | -1.830** | -0.636** |
|  | (0.563) | (0.195) |
| Treatment FD | -2.171** | -0.753** |
|  | (1.040) | (0.361) |
| Time trend | 0.163** | 0.056** |
|  | (0.019) | (0.006) |
| Individual initial decision | 0.178* | 0.062* |
|  | (0.095) | (0.033) |
| Intercept | -8.229 |  |
|  | (5.148) |  |
| Log-likelihood | -2129.767 |  |
| Wald test for model significance | $\chi2$ (63) = 902.19 | p-value<0.001 |
| LR test for CRE dynamic Tobit without controls | $\chi2$ (21) = 143.09 | p-value<0.001 |
| LR test for standard RE static Tobit | $\chi2$ (59) = 278.01 | p-value<0.001 |
| Number of observations | 1976 |  |
| Number of individuals | 104 |  |
| Uncensored observations | 672 |  |
| Left-censored observations | 26 |  |
| Right-censored observations | 1278 |  |

Notes: Standard errors are given in brackets. Significance level:

*10%

**5%.

effect (although not as strong) due to the first decision as underlined by [47]. Table 4 presents estimation results. We compared our CRE dynamic Tobit model to the static Tobit model with standard random effects by using a likelihood-ratio $\chi^2$ test. The results showed unambiguously that the static model was dominated by our model for the whole sample. We also ran the CRE dynamic Tobit model without control variables ("Decision-making time" and "Time trend"). The signs of the coefficients of our main variables remained unchanged. Moreover, the likelihood-ratio test (which was a $\chi^2$ distributed under the null) showed that our model was strongly preferred. We report the estimated coefficients and the corresponding partial effects of explanatory variables on the expected value of individual extraction (given that it is censored at 0 and 10). Note that this quantity is defined by $E(y_{it}) = Pr(0 \leq y_{it} \leq 10) * E(y_{it}|0 \leq y_{it} \leq 10) + 10 * Pr(y_{it} > 10)$, where $E(y_{it}|0 \leq y_{it} \leq 10)$ is the expected value of the dependent variable when it is truncated at 0 and 10. The estimated partial effects are globally consistent with the estimated coefficients as the signs and the significance are very similar. We find that both treatments where disclosure of decisions is on a voluntary basis; specifically the VD and FD treatments, have a significant negative impact on the amount extracted in the game (however, the two coefficients are not different from each other, $\chi^2 = 0.10$, p-value = 0.755). This confirms what can be seen in Fig 2, and also confirms the non-parametric tests reported above. The estimation further shows that individual decisions are strongly related to what the players observed from the group during the previous round and that there is a natural tendency towards higher extraction as time elapses. Estimates also reveal that extracting less from the common resource seems to be based on a more cognitive decision-making process since the shorter the decision-time, the higher the extraction [48]. This is consistent with the study of [49] which found that faster subjects more often choose the option with the highest payoff for themselves. Finally, the estimation shows that with all else equal, the first extraction decision in the game matters. This decision was taken without any prior knowledge about the behavior of the group as it took place just after the reading of the instructions for the experiment. [47] also observed this phenomenon. Hence, even if a mechanism tries to influence the dynamics of the individual's decision process, the individual's initial intention remains a strong anchor [50].

## Information sharing effect

Fig 3 provides several useful curves to understand what happened in the two voluntary disclosure treatments. The VD treatment is on the left graph and the FD treatment is on the right graph. First, there is the average of the non-disclosed extractions (dotted line and triangle marker facing down). Second, there is the average of the disclosed extractions (dashed line and triangle marker). In the VD treatment, this average corresponds exactly to the curve of the displayed extractions. For the FD treatment we have added the average of the displayed extractions, which in this treatment can be different from the actual extractions of the players (dash-dot line with pentagon marker). Finally, we report the average extraction (plain line with star marker), i.e. the same curves as in Fig 2 for these treatments.

Clearly, the players who decided to disclose their decision extracted less than the others. This holds in both treatments (Wilcoxon test p-value < .001 in both treatments), with a larger difference for VD than for FD, however. This supports conjectures 2.1 and 3.1. In the VD treatment, we find that the overall average of the extractions and the average of the disclosed extractions (disclosed value) are very close and follow the same trajectory. This is consistent with the expected effects of social information. These being: imitation; convergence of decisions with observed decisions and creation of a norm within the group [10–12, 19, 38–40]. Conversely, in the FD treatment the average of the displayed extractions differs greatly from

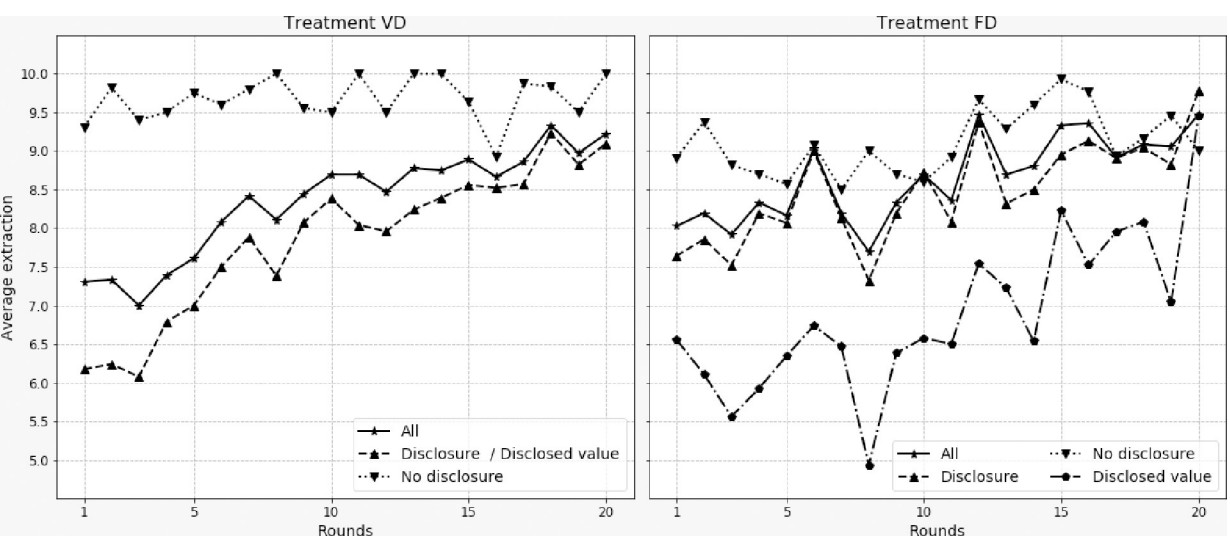

**Fig 3. Evolution of average extractions depending on whether or not they were disclosed.**

the average of the actual extractions of the group, thus being in line with conjecture 3.2. The average extraction curve for players who did not disclose their choices is higher in the VD treatment than in the FD treatment. The difference comes from the fact that in the VD treatment the least cooperative individuals did not disclose their choices, whereas in the FD treatment they disclosed an extraction value, even if it did not correspond to their actual extraction.

Fig 4 helps to further understand the FD treatment. The graph shows three new curves in addition to the curve of the average extractions of players who chose not to disclose their

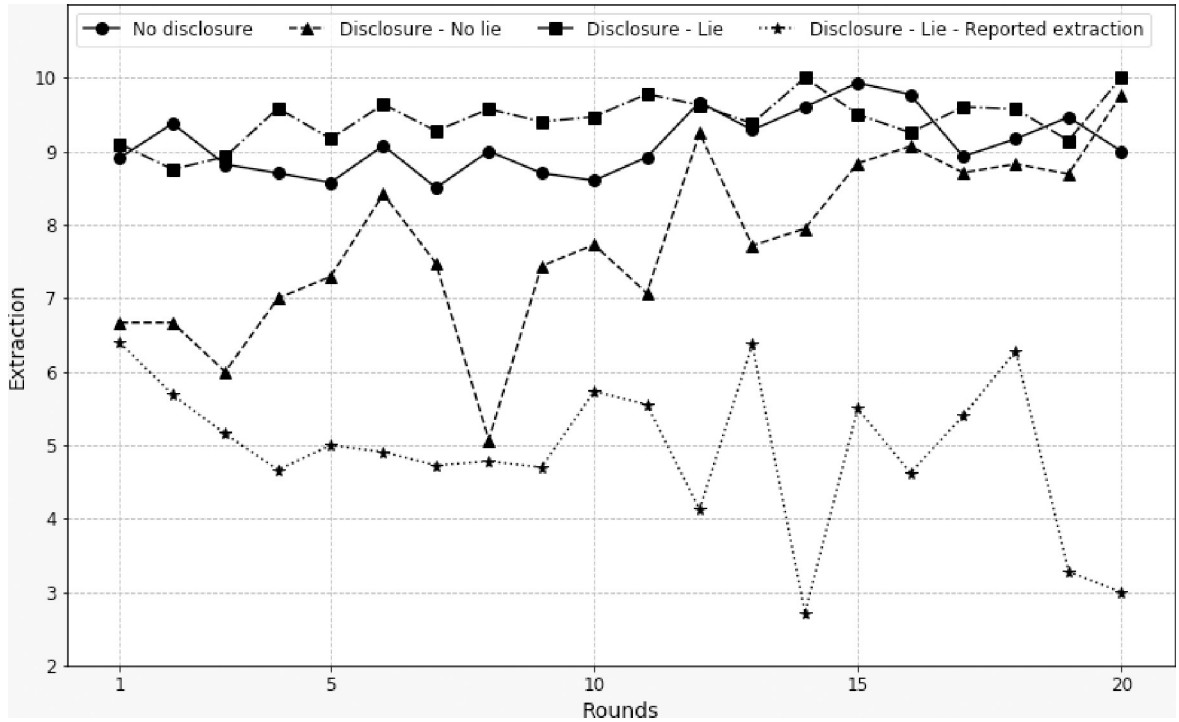

**Fig 4. Actual extractions in FD for players who did not disclose their decisions, for players who disclosed their actual extraction, and for players who disclosed an amount different from the actual one.** For the last case the disclosed amount is also shown.

extractions. The first curve corresponds to the average of the extractions of players who disclosed their decision and did not lie in the value they entered (dotted line with triangle marker). The second curve shows the average of the extractions of players who disclosed their decision but entered a value different from their actual extraction (dash-dot line with square marker). Finally, the third curve shows the average of the values entered by the latter players (small dotted line with star marker). It is clear that players who lied about the amount they reported extracted far more than those who disclosed their actual extraction (Wilcoxon p-value < .001) and also more than those who refused to make their decision public (Wilcoxon p-value = .010). However, as can be seen with the curve of the values they entered, these players seem to have quickly understood that the social optimum was to extract 5 units. The freedom given to the players to decide for themselves the extraction that would be disclosed favored the emergence of a strategic behavior consisting in reporting an extraction close to the social optimum. They sent a false signal, expecting others to decrease their extraction in order to increase their individual profit. After the tenth round, when the average extraction starts to increase, one can observe a disclosed extraction by lying players that decreases. A simple linear regression (individual reported extractions on rounds) gives a coefficient value -0.08 for rounds 1 to 10 and to -0.14 for rounds 11 to 20, which are not statistically different at the 5% level. We have not explored the reason why these participants report an extraction level lower than the socially optimal value. The assumption we can make is that it is a way to slow down the increase towards the maximum extraction. In order to identify the effects of specific explanatory variables (in particular those related to information sharing), estimations were carried out treatment-by-treatment. Table 5 presents the estimation results of the same model (dynamic Tobit with correlated random effects) for the MD, VD, and FD treatments respectively. As in the case of the whole sample, we performed an LR test to compare the models with and without correlated random effects (i.e. null hypothesis $\alpha_1 = \gamma = 0$) for each of the three treatments. The result was unambiguously in favor of the CRE dynamic Tobit model (test statistic was 61.051, 93.543, and 90.688 for the MD, VD and FD treatments respectively). In addition, test results for every treatment were also favorable to our model when it was compared to either the static Tobit model with standard random effects or the CRE dynamic Tobit model without control variables (see Table 5). As the estimated partial effects are consistent with the estimated coefficients (i.e. they have similar signs and significance levels), we can rely on any of the two sets of coefficients to interpret the results. Table 5 provides estimation results for the MD treatment in the first column. By definition, the set of explanatory variables does not contain any factors related to the voluntary sharing mechanism. The estimated coefficients are therefore similar to the case of the whole sample presented in Table 4.

The models estimated for the VD and FD treatments include additional variables linked to the voluntary disclosure mechanism. More precisely, for the VD treatment we added two dummy variables to indicate whether or not players consented to disclose their extractions in the current and in the previous round ("Information sharing, current round" and "Information sharing, previous round"). We also added the number of members in the group who chose to disclose their individual decision. In the FD treatment; as players could report an extraction that was different from the actual one, there were three possible situations, each of them corresponding to a dummy variable: (i) the players refused to disclose their decision (the reference); (ii) they consented but reported an extraction that was different from the actual one ("Information sharing & lying") and (iii) they consented and reported their actual extraction ("Information sharing & non-lying"). We added the present and the past value for the latter two dummies. It should be noted that including whether or not extractions were disclosed might have created an estimation bias. Indeed, individuals could simultaneously make multiple decisions about (i) their extraction; (ii) their choice on whether or not to disclose their

**Table 5. Estimation results by treatment, using the CRE dynamic Tobit model with individual extraction as the dependent variable.**

| | MD | | VD | | FD | |
|---|---|---|---|---|---|---|
| Variable | Coefficient | Partial effect | Coefficient | Partial effect | Coefficient | Partial effect |
| | (Std. Err.) | (Std.Err.) | (Std. Err.) | (Std.Err.) | (Std. Err.) | (Std.Err.) |
| Individual past decision | 0.260* | 0.058* | 0.217** | 0.090** | -0.333** | -0.131** |
| | (0.140) | (0.031) | (0.106) | (0.044) | (0.133) | (0.053) |
| Group past decision | 0.189** | 0.042** | 0.065* | 0.027* | 0.178** | 0.070** |
| | (0.068) | (0.015) | (0.037) | (0.015) | (0.061) | (0.024) |
| Decision-making time | -0.049** | -0.011** | -0.059** | -0.024** | -0.073** | -0.029** |
| | (0.018) | (0.004) | (0.014) | (0.006) | (0.021) | (0.008) |
| Time trend | 0.133** | 0.030** | 0.175** | 0.072** | 0.139** | 0.055** |
| | (0.044) | (0.010) | (0.029) | (0.012) | (0.056) | (0.022) |
| Individual initial decision | 0.059 | 0.013 | 0.476** | 0.197** | -0.119 | -0.047 |
| | (0.253) | (0.057) | (0.141) | (0.058) | (0.126) | (0.050) |
| Information sharing, current round | | | 4.629 | 1.915 | | |
| | | | (3.384) | (1.396) | | |
| Information sharing, previous round | | | -3.382** | -1.399** | | |
| | | | (1.549) | (0.638) | | |
| Information sharing, #members in the group | | | 0.013 | 0.005 | -0.180 | -0.071 |
| | | | (0.247) | (0.102) | (0.289) | (0.114) |
| Information sharing & lying, current round | | | | | 4.530 | 1.791 |
| | | | | | (3.855) | (1.523) |
| Information sharing & non-lying, current round | | | | | 8.347 | 3.301 |
| | | | | | (5.276) | (2.082) |
| Information sharing & lying, previous round | | | | | -2.188 | -0.865 |
| | | | | | (1.748) | (0.690) |
| Information sharing & non-lying, previous round | | | | | -4.731* | -1.871* |
| | | | | | (2.694) | (1.063) |
| Intercept | -10.860 | | -6.737** | | -14.047** | |
| | (14.438) | | (1.596) | | (4.576) | |
| Log-likelihood | -470.06 | | -783.55 | | -823.01 | |
| Wald test for model significance | X2 (23) = 165.15 | p < .001 | X2 (27) = 534.49 | p < .001 | X2 (30) = 345.13 | p < .001 |
| LR test for CRE dynamic Tobit without controls | X2 (2) = 20.15 | p < .001 | X2 (2) = 70.16 | p < .001 | X2 (2) = 51.23 | p < .001 |
| LR test for standard RE static Tobit | X2 (21) = 67.12 | p < .001 | X2 (21) = 94.38 | p < .001 | X2 (20) = 92.02 | p < .001 |
| Number of observations | 608 | | 684 | | 684 | |
| Number of individuals | 32 | | 36 | | 36 | |
| Uncensored observations | 133 | | 269 | | 270 | |
| Left-censored observations | 2 | | 4 | | 20 | |
| Right-censored observations | 473 | | 411 | | 394 | |

Notes: Standard errors are given in brackets. Significance level:

*10%

**5%.

decision and (iii) the amount they reported. This phenomenon led us to consider the corresponding explanatory variables as endogenous regressors (i.e. the variable "Information sharing, current period" for the VD treatment, and the variables "Information sharing & lying, current round" and "Information sharing & non-lying, current round" for the FD treatment). For this purpose, we applied the control function approach proposed by [46], which is

particularly suitable for nonlinear models such as our Tobit model with correlated random effects. Table 5 provides estimation results that account for this endogeneity bias. Based on a robust t-test (also proposed by [46]), we found that the exogeneity of these regressors could not be held even if their coefficients were not statistically significant. In other words, an estimation assuming the exogeneity of these regressors could lead to misinterpretation.

The control function approach of [46], consisting of a two-step estimation, is relatively simple to implement. At the first step, a probit model for the endogenous regressor is estimated in order to obtain a generalized residual. The second step corresponds to the estimation of the usual nonlinear model (i.e. Tobit model with correlated random effects) with the previously computed generalized residuals as an additional regressor. See [46] for more computational details. Lastly, we performed a robust t-test for the significance of these generalized residuals. For the VD treatment, the t-statistic was -2.08, while for the FD treatment, the t-statistic was -1.20 for the first generalized residuals (corresponding to "Information sharing & lying, current round") and -1.97 for the second generalized residuals ("Information sharing & non-lying, current round"). This result implies the significance of generalized residuals in the non-linear regressions, thereby supporting the control for endogeneity of information sharing when using data in the VD and FD treatments.

In the VD treatment, the model estimates confirm that the player's decision in the current round was strongly influenced by the player's decision in the first round of the game and by their decision in the previous round. The anchoring effect observed in Table 4 is ultimately significant only in this treatment. For us this can be interpreted by the fact that there is less noise in the information available to players in the VD treatment, as players can trust the information and easily identify good will. The total extraction of the group in the previous round also exerted a strong influence on the player's current extraction decision. Moreover, estimates confirm that a player who decided to make their extraction public in the previous round extracted less in the current round than an individual who preferred to keep their extraction private in the previous round (supporting conjecture 2.1). In the FD treatment, the extraction of the player in the previous round had a negative impact on their extraction in the current round, and the extraction of the player in the first round of the game was not a significant variable in the model. Our interpretation is that the social information in this treatment is less selective and has a strategic dimension, which seems to result in players fluctuating more in their extraction decisions. The estimates also tell us that what the player did in the previous round had more influence than what they intended to do in the current round. Thus, the information sharing variables (yes with a lie or yes without a lie) of the current round are not significant compared to the reference variable (no information sharing). A player who decided to make their "true" extraction public in the previous round (i.e. without a lie) extracted less in the current round than a player who preferred to keep their extraction private or who made it public but displayed a "false" value (i.e. lied). Thus, in both voluntary sharing treatments, it can be said that the individuals who contribute to better management of the common resource are those who are willing to make their extractions public with complete honesty and transparency.

## Conclusion

The management of common resources is an everyday universal challenge on both a large scale; such as with the oceans or the atmosphere, and also in local situations; such as the use of a borehole shared by a small community. Many factors contribute to improving the management of these resources, as was shown by Nobel Prize winner Elinor Ostrom in numerous articles. Among these factors there is information about the actions of other users of the resource, which is commonly referred to as social information. The effects of this social information are

mixed. On one hand, it stimulates cooperative actions, as users may feel obliged to show signs of cooperation [5]. Moreover, it provides users with the opportunity to send a signal about the expected cooperation within the group and the social norm that seems appropriate [16, 19]. On the other hand, however, it is likely to spotlight the least cooperative users. Those users; due to the structure of the social dilemma, have higher earnings than cooperative users as the exploitation of the resource generates individual profits. Therefore, if users tend to imitate the best performance, over-exploitation of the resource is inevitable and can be accelerated by social information [8].

Providing information about the actions taken by all the users in the group is therefore not necessarily beneficial. In this article we proposed setting up a system of voluntary sharing of information, in order to keep the beneficial effects of disseminating information (signal) and mitigate its negative effects (imitation of the most selfish), and therefore to favor the emergence of upward social information [10]. We experimentally tested two voluntary information sharing mechanisms. In the first one, the players were only invited to indicate whether they agreed to publicize their level of extraction. In the second one, the players were free to decide whether or not to disclose their level of extraction, but they were additionally responsible for reporting the amount that was made public. We compared these treatments with a benchmark treatment in which individual decisions were automatically and mandatorily disclosed. We could have used a benchmark with no social information at all, i.e. with only aggregate information on the total amount extracted by the group. This may be a limitation of this study, but our main concern was to test a mechanism that allows for self-selection of the social information provided to users of a common resource. For the two voluntary disclosure mechanisms, the data exhibited a lower average extraction compared to the system with automatic and mandatory disclosure of decisions and therefore reduced the phenomenon of over-exploitation of the resource. The main reason is that voluntary disclosure allows selection in the social information disseminated to players. The more cooperative players make their actions public, while less cooperative players keep this information private, so as not to have a negative influence on others and thus accelerate the tragedy of the commons. However, while the voluntary disclosure mechanism leaves players free to set the reported amount extracted, less cooperative players exploit the strategic dimension. Those players do not hesitate to lie in an attempt to gain more from the common-pool resource. Ultimately, this is likely to undermine the beneficial effect of the mechanism by breaking trust in the group and preventing the formation of a social norm.

[21] showed that the voluntary disclosure mechanism improves the average contribution to the production of a public good. Our study showed that the benefits of this mechanism are still valid in an extractive common resource context. Further investigation is however required to better understand the ins and outs of voluntary information disclosure mechanisms. For example, [21] tested the mechanism in small groups like we did (group size of 5 and 4, respectively). It would be interesting to test it in larger groups where it would be more difficult to influence the norm or to infer the extraction levels that others do not disclose. Furthermore, information was shared anonymously, so the effects of the reputation mechanism when anonymity is lifted may need to be studied. This could increase the motivation of individuals to signal themselves as cooperative, and also limit the strategic behavior of displaying a value lower than the actual extraction. In the same vein, identifying the type of players upstream of the game (unconditional cooperator, conditional cooperator, free-rider or other); with a procedure inspired by those proposed by [20] and [51], would make it possible to learn more about individual behaviour in the game according to the voluntary sharing mechanism put in place. Finally, it would be interesting to test the voluntary information-sharing mechanism in

a dynamic environment, where the resource is constantly evolving. It is a more complex framework but also closer to reality for common resources [52].

The development of new technologies and of the Internet of Things (IoT) greatly facilitates the sharing and the disclosure of information through smart devices. As a result of this, consideration can be given to new mechanisms for managing common-pool resources. Smart grids and smart meters are examples of devices that allow this purpose to be pursued. For instance, in the electricity sector regulators are able to monitor grid traffic in real time and take appropriate actions to reduce stress on the grid in peak hours. At the same time, users can receive information; such as electricity pricing, through variable pricing on a real time basis and are better incentivized to manage and adjust their energy consumption [53]. Using smart devices to manage common resources does however come up against the problem of the social acceptability of the automatic and mandatory collection of individual data. Adopting voluntary based data collection could help to solve part of this problem.

The efficient management of a common-pool resource at a reasonable cost is a serious challenge for decision-makers. The voluntary dimension of providing information about resource extraction can be a useful tool in this direction. If the regulator only installs smart meters for voluntary users, it reduces costs compared to a generalized installation. In addition, if the regulator asks users to self-report their extractions, it eliminates metering installation and operating costs. The cost in this case is for the user to self-report. However, audits need to accompany self-reporting systems and partial installations of meters. The regulator must therefore calculate the trade-off between the costs of installing meters and the costs of auditing unmetered users. When users self-report their extractions, the number of audits should be higher, as some users will declare amounts that are different from their actual extractions. In California for instance, the law on sustainable groundwater management; adopted by the State in 2004, stipulates that the board requires annual extraction reports and that metering may be required to acquire the data. The board can then issue orders to acquire the information that is needed. The full text can be found online at https://mavensnotebook.com/2020/01/23/sgma-implementation-groundwater-sustainability-evaluation-and-state-water-board-intervention/ and https://www.waterboards.ca.gov/water_issues/programs/gmp/intervention.html.

The dissemination of extraction data for social information purposes represents a step towards voluntary provision, i.e. the case data collected by the regulator is revealed to end-users. As many authors have shown with laboratory and field experiments, social information and the following social comparison (see also [54–56] for research on social comparison nudges), lead users to adopt more cooperative behaviors. With a voluntary disclosure mechanism, the dissemination of social information becomes possible, and since it is selective, its efficiency is increased; as shown by [21] and by us in this paper. The free information disclosure mechanism we have proposed in this article is even more likely to foster social acceptability. However, as we have seen, it introduces opportunistic and strategic behaviors which may offset the benefits of social information in the long run and on a larger population scale. To address this problem the mechanism could be accompanied by audit systems similar to those for tax returns. This would increase the cost of the mechanism but is also likely to further increase its effectiveness. This is something that should be examined and tested in further investigations.

## Supporting information

**S1 Appendix.**
(DOCX)

## Acknowledgments

The authors are grateful to the Experimental Economics Laboratory in Montpellier for technical support, to P. Biggins for his careful reading and comments, and to participants at the AEA-ASSA conference (San Diego January 3–5, 2020), the Public Economic Theory (PET) conference (Strasbourg, July 8–11, 2019), the Environment for Development (EfD) Initiative (11th Annual Meeting, Hanoi, November 2–5, 2018), the IBCCS Workshop (Hanoi, July 19, 2018), the BETA seminar (Nancy, March 19, 2019) and the ASFEE (French Association of Experimental Economics) conference (Nice, June, 14–15, 2018). Finally, the authors would like to thank the reviewers for their useful comments and suggestions, which have improved the quality of the manuscript.

## Author Contributions

**Conceptualization:** Dimitri Dubois, Stefano Farolfi, Juliette Rouchier.

**Formal analysis:** Dimitri Dubois, Phu Nguyen-Van.

**Methodology:** Dimitri Dubois, Stefano Farolfi, Juliette Rouchier.

**Software:** Dimitri Dubois.

**Writing – original draft:** Dimitri Dubois, Stefano Farolfi, Phu Nguyen-Van, Juliette Rouchier.

**Writing – review & editing:** Dimitri Dubois, Stefano Farolfi, Phu Nguyen-Van, Juliette Rouchier.

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
