## [Decision Letter · Decision Letter 0]

31 Mar 2020

PONE-D-20-03051

Contrasting effects of information sharing on CPR extraction behaviour: experimental findings

PLOS ONE

Dear Dr. DUBOIS,

Thank you for submitting your manuscript to PLOS ONE. After careful consideration, we feel that it has merit but does not fully meet PLOS ONE’s publication criteria as it currently stands. Therefore, we invite you to submit a revised version of the manuscript that addresses the points raised during the review process.

We would appreciate receiving your revised manuscript by May 15 2020 11:59PM. To enhance the reproducibility of your results, we recommend that if applicable you deposit your laboratory protocols in protocols.io, where a protocol can be assigned its own identifier (DOI) such that it can be cited independently in the future. For instructions see: http://journals.plos.org/plosone/s/submission-guidelines#loc-laboratory-protocols

We look forward to receiving your revised manuscript.

Kind regards,

Valerio Capraro

Academic Editor

PLOS ONE

Journal Requirements:

2. Please improving statistical reporting and refer to p-values as "p<.001" instead of "p=0". Our statistical reporting guidelines are available at https://journals.plos.org/plosone/s/submission-guidelines#loc-statistical-reporting

Additional Editor Comments (if provided):

I have now collected four reviews from four experts in the field. The reviews are somehow split, with one recommending rejection and three recommending major revision. Therefore, I would like to invite you to revise your work following the reviewers' comments. Needless to say that all comments must be addressed. Besides the reviewers' comments, I would like to ask you to avoid using acronyms in the abstract (they minimise impact).

I am looking forward for the revision.

Reviewers' comments:

Reviewer's Responses to Questions

**Comments to the Author**

1. Is the manuscript technically sound, and do the data support the conclusions?

Reviewer #1: Partly

Reviewer #2: Partly

Reviewer #3: Partly

Reviewer #4: Partly

2. Has the statistical analysis been performed appropriately and rigorously? 

Reviewer #1: No

Reviewer #2: Yes

Reviewer #3: Yes

Reviewer #4: Yes

3. Have the authors made all data underlying the findings in their manuscript fully available?

Reviewer #1: Yes

Reviewer #2: No

Reviewer #3: No

Reviewer #4: Yes

4. Is the manuscript presented in an intelligible fashion and written in standard English?

Reviewer #1: Yes

Reviewer #2: Yes

Reviewer #3: Yes

Reviewer #4: Yes

5. Review Comments to the Author

Reviewer #1: The paper studies the effect of extraction disclosure in a common pool resource game. Based on a controlled lab experiment, the authors introduce three different treatments wherein they test the effect of a mandatory disclosure policy, a voluntary one and a policy characterized by the freedom to disclose any amount. They find that a voluntary policy would help to reduce extraction compared to a mandatory one. However the freedom to disclose appear to be counter-productive.

Although the research question is interesting, but I have some concerns about the robustness of the approach and the contribution of the paper appears to be small.

Comments:

- The authors present a series of example to motivate the paper but they coul explain why they have decided to introduce these interventions in a CPR game setting. There is mixed evidence for the external validity of CPR experimental studies and behavior in the experiments sometimes does not match observed behavior outside the experiment. It would be interesting to see some motivation for the choice of the game.

- The section dedicated to the conjectures is not very informative. It does not help to understand how subjects are expected to behave in each treatment and, more importantly, why. Especially it would be interesting to discuss what are the possible mechanisms that might explain the positive effects of the main treatments relative to a baseline CPR game. Currently, the paper is quick on this and just evoke the formation of social norms. What about dishonest behavior by free-riders in the VD treatment? Having such a discussion could improve the added value and contribution of the paper.

- There are other (potentially relevant) papers on CPR game with communication or social interactions prior to making decisions, see for example, Cardenas et al (2000), Rodriguez-Sickert (2008) or Andries et al (2011) for a survey of literature. How does your paper relate to this strand of the literature?

- Overall, the authors should spend more effort in explaining why and how their paper extends the existing literature on CPR experiments.

- In several occasions, the authors write that in social dilemmas “most players are conditional cooperators” and cite Fischbacher et al (2001). However, this is not true, Fischbacher et al found that on average about 50% of subjects are found to be conditional cooperators and recent studies have mostly confirmed this figure.

- I am not convinced by the analysis in Section 3:

o There is no non-parametric analysis of the average behaviours in the treatments. Given the small size of the groups and the small variation of averages between groups, I suspect there is no significant differences but the authors should discuss it.

o Also median and/or distribution analysis could be performed here in order to give insights on the potential differences.

o In the regression analysis, it is not clear to me why they use a dynamic panel model with previous extraction and initial extraction (that is not really defined by the way) and not a simple random effect tobit or OLS with no dynamics. Indeed, including the lagged dependent variable may lead to estimation problems due to endogeneity issues. Nothing specific is said about this issue but the error term can't be statistically independent of y(t-1).

o Also it would be interesting to run an estimate without the demographic to see how the effect is dependent on the controls.

Reviewer #2: The paper studies the relative efficiency of three information disclosure mechanisms on common pool resources (CPR) problems: Mandatory disclosure, voluntary disclosure and free disclosure. For this purpose, the authors run an experiment on CPR where groups of 4 individuals decide on the number of tokens (from 1 to 10) they extract, taking into account that their payoffs increase with the tokens they get, but also that there is a negative externality to the group increasing with the total number of tokens extracted in the group. Although the symmetric equilibrium of the resulting game would imply to extract 5 tokens each, the average number of extractions during the 20 rounds is between 7 and 10. These extractions are higher when a mandatory disclosure on the extractions is implemented than when voluntary mechanisms are considered, although the number of extractions increase and converge to the maximum along the experiment in all treatments. However, the results show that there is a significant number of ‘cooperative’ people that decided to disclose their decisions, which is higher in voluntary than in free disclosing treatment, since the latter tend to exploit the disclose information strategically.

Although the results of the are somehow expected, the experiment is appealing, apparently well-executed and the analysis well-performed, which are the main items for publication in PLoS ONE. However, I also believe that there are several issues that should be considered in the revision. See my main comments below:

Comments

1) There is a paper on CPR published in PLoS ONE (Lacomba et al. 2017) that shows that there is a significant number of cooperators (which decide to follow a sustainable rule of extractions) independently on the disclosure of information about the actions of other players. However, the (mandatory) disclose of information either accelerates the non-cooperative behavior or lead to a more egalitarian share of the resources. These two dynamics are highly dependent on the first choice of the players. This story is also found in the paper under revision, where the initial decisions have a positive and significant impact in the voluntary contribution treatment. Nevertheless, these ideas are not exploited enough in the paper. I suggest to study all these issues in depth.

2) Linked to the previous comment, I miss a baseline treatment where there is no disclosure of information at all. Would all subjects behave as free-riders or even in this case there would be a significant proportion on cooperators?

3) From the average results in Figure 2 the people that follow the optimal strategy (taken just 5 tokens) cannot be appreciated. It would be interesting an analysis on the percentage of people who follow the ‘optimal strategy’ in all treatments. I have the impression that subjects are not capable of deducing their optimal strategy from the payoff function in the experiment. Maybe they should have received some examples in the instructions to clarify it. I think that the instructions of the experiment should be appended in the paper to shed light on this issue.

4) The results seem to be highly dependent on the number of people in the groups. For two individuals (as in the paper mentioned in comment #1) free-riders are immediately detected with the disclosure of information. With four subjects, free riders (and information manipulators) can be identified along the experiment and this explains the convergence of voluntary and free disclosure trends. An analysis of the sensibility of results to the number of subjects would be performed or at least mentioned as a possible limitation.

5) It is not clear to me if the students were economically incentivized or not (for instance Conjecture 1 states “Even without a specific monetary incentive”). The experimental economics literature is very critical with non-monetary-incentivized experiments. Therefore, this issue should be also clarified. In general, I miss more details on the experimental procedures, as well as the instructions.

6) The paper run panel data regressions to investigate on the individual decisions on extractions. I would devote a particular Section for the econometric modeling. The models and the variables used should be properly justified, as well as the employed method (a tobit model with correlated random effects). Note that dynamic panel data models, by construction, present endogeneity problems that should be taken into account (maybe through GMM estimation). The group effects have to be treated as well, since every group has different interactions. All the tables can be gathered in a single table so as estimates can be easily compared and statistics on the model performance should be also displayed. As a matter of fact, the interpretation of the results deserves further explanations.

7) It would be also interesting to analyze gender issues in relation to free-riding, voluntary discussing and manipulation on reported information in free disclosure treatment. Are any significant differences between men and women?

8) The paper states that the free disclosure mechanism seems to be less efficient than voluntary disclosure, although less expensive since it does not require an agency for data collection. Nevertheless, it would require an external audit to manage with the strategic manipulation of data. An alternative solution would be the implementation of ‘strategy-proof’ mechanisms that incentivize truth-reporting. I suggest to look for alternative payoff functions where the penalty be less sensitive to the deviations from the optimal strategy (maybe in terms of N times de median extraction) or where an extra penalty for randomly detection for liars be introduced.

References

Lacomba, J.A., Lagos, F., Perote, J. (2017). The Lazarillo's game: Sharing resources with asymmetric conditions. PLOS ONE 12/7, e0180421.

Reviewer #3: Review of: Contrasting effects of information sharing on CPR extraction behaviour: experimental findings

The paper presents results from a behavioral lab experiment designed to test the effect of different information sharing regimes on the extraction levels in a repeated common-pool resource (CPR) game played by groups of four participants. Results show that if participants are given a choice of whether or not to disclose their extraction level in each round, extraction levels are closer to the optimal level at the start as compared to the automatic, full disclosure condition, but then converge to inefficient, maximum extraction levels over time.

I like the idea behind this experiment: If cooperation (optimum level extraction) is driven by descriptive norms, giving participants the choice of whether or not to disclose their individual extraction levels may lead to higher cooperation because free-riders will be less likely to disclose their full extraction levels. The results of this experiment thus inform us that implementing voluntary disclosure mechanisms may not be sufficient to promote sustainable cooperation in a four-person, repeated CPR game. This is an important insight. However, as the paper is currently written, the papers’ contribution to our knowledge may remain unclear for inattentive readers. Here are my suggestions on how the clarity of this paper could be improved (in order of appearance in the paper):

Abstract: The last sentence is not very informative. The authors repeat this sentence at the end of the introduction, but there it is not informative either. In essence I believe there is not much the authors can say about the two treatment conditions except that in both participants behavior converges to maximum extraction. I’d therefore replace this sentence with one that better reflects the actual results of this study.

At the bottom of page 3 the authors briefly refer to previous literature on factors that promote cooperation in CPR games and mention communication, reputation and information as important factors. Somewhat later it becomes clear that the factor the authors focus on is information. However, communication and reputation also involve information. The paper would benefit if the authors were more specific about what they mean by information already at this point in the paper. The paper might also benefit from a somewhat longer review (one or two paragraphs) of these mechanisms in the CPR context.

As mentioned already above, the main theoretical argument implicit in the introduction is that pro-social participants (i.e. participants with other-regarding preferences) would extract less and disclose their true extraction levels more than free-riders and this would lead to average extraction levels closer to the optimum due to descriptive norms. But what is unclear in this argument is why pro-socials will be more likely to disclose their true extraction levels? Discloser as such does not produce any benefits for others, only extraction levels do. So why is truthful information sharing in this case a pro-social act? Only because disclosure is costly is not sufficient an argument to infer pro-social preferences in those who disclose. Clarifying this point is all the more important as it is not costly at all to disclose one’s extraction level in the experiment.

In the first paragraph on page 5 the authors talk about framing to say that it may matter whether the social dilemma is a give-some (e.g. PGG) or take-some (CPR) dilemma. I think they should acknowledge some relevant literature on this topic (e.g., van Dijk and Wilke, 2000; Gächter et al, 2017). In this same paragraph the authors mention a study by Kreitmair (2015), which is most similar to what they do in this paper. But then they never get back to Kreitmair (2015) in the discussion. I think they should discuss how their results compare to Kreitmair (2015).

In the second paragraph on page 5 the authors describe their experimental conditions. The description of the free disclosure condition (FD) should be improved. To what do players agree? And they declare themselves as what? I think it is only a matter of wording. And once these terms are introduced, the authors should stick to using the same throughout the paper (e.g. use mandatory rather than compulsory, etc.).

In the same paragraph the authors write: “The FD mechanism is free of charge for the regulatory agency because the information is provided by the actors themselves.” Does it matter for your argument that it might be the most costly measure for consumers? They need to go online and enter a value each period.

On page 7 the experimental conditions are explained once more. In the paragraph describing the voluntary disclosure treatment the authors write: ”Specifically, it was anonymously displayed in the summary screen…” What do the authors mean by that?

The experimental design and procedure (section 2.2) are insufficiently described. Information on the number of sessions, participants per session, the randomization procedure, length of sessions, how much participants earned on average, etc. are missing. Moreover, it is standard in experimental social science to add a version of the instructions (translated into English) in the online appendix. The authors need to add this information to allow for a better assessment of their use of methods. Also, I was wondering whether they could add participants’ average age in footnote 7.

Table 2 should also contain the number of cases and standard deviations.

First line on page 11 should say Figure 2 (not Figure 1), no? Same is probably true for last sentence on page 12.

I’d prefer if the authors compiled all four regression models in one table. By putting the standard errors below the coefficient estimates it should be possible. One table will facilitate the comparison of effects across different models. Moreover, information about the target variable should be placed in the title of the tables; it is currently hidden in the table notes.

In Table 3 the effect of treatment FD is larger than the effect of VD. This seems inconsistent with what Figure 2 shows. The authors should maybe check the labels in Table 3 once more.

In Figure 4 I did not get what the dotted line with the star symbol stands for. It does not seem to be explained in the text either.

Note that information sharing can also be included in the model fitted to all data. This can be done by creating a dummy for whether a participant is in the mandatory disclosure condition (0) or in one of the voluntary disclosure conditions (1) and interacting this dummy with the information sharing variables that are currently only in Tables 5 and 6. Relatedly I was wondering why the authors include a variable for “information sharing in current period”. Do participants first decide whether to share the info and then the amount they want to extract? If so, that was not clear from the description of the experimental design and procedures and therefore should be added there.

The discussion section is very brief and superficial. Apart from relating their results to the results by Kreitmair (2015), the authors could discuss some obvious limitations of their study and point out directions for future research. As far as I can see, the main limitations of the study are: (1) there is no control condition without information provision, which would correspond to the standard nowadays (most people do not know how much their neighbors extract or what the average in their neighborhood or their village/ city is). (2) Even if free-riders did not disclose their extraction level, given the small groups (n =4) participants could infer the number of free riders from average or even own earnings in a round. Finally, I was wondering whether the authors can say a bit more about the extent of conditional cooperation based on their results. By expanding this discussion they could tie their paper back to the start where they talk about social norms and conditional cooperation.

Finally note that the authors’ statement in the very last paragraph that “Observations from our experiment showed that a mechanism based on voluntary sharing is effective,…” is not justified given the results. Eventually, all groups reach close to maximum extraction level. The authors should adjust their conclusions accordingly.

I will be happy to have another look at the revised version but the authors should seriously consider all the points I make above in their revision.

References

Gächter, S., Kölle, F., & Quercia, S. Reciprocity and the tragedies of maintaining and providing the commons. Nature Human Behaviour, 1, 650-656.

van Dijk, E., & Wilke, H. (2000). Decision-induced focusing in social dilemmas: Give-some, keep-some, take-some, and leave-some dilemmas. Journal of Personality and Social Psychology, 78, 92-104.

Reviewer #4: The main question in this research is compelling. However, there are several issues and biases that authors need to handle to overcome the shortcomings of this paper. It appears that authors have taken several things in a very light manner without paying enough attention. The current experiments could not fully convince the readers how the mechanism of voluntary information disclosure can be implemented in real life. However, it is a new issue that authors are trying to handle, but it is crucial to be clear what the authors can claim and cannot form the current experiments. It is advisable to be clear on the part “scope of experiments” and to state the limitations and future work if something is out of scope. Therefore, to publish this paper, the authors need to handle the significant issues raised seriously. Moreover, authors should also keep in mind that this work is going to be published in the multidisciplinary journal. Therefore, policy implication needs to clear cut for the general readers.

6. PLOS authors have the option to publish the peer review history of their article (what does this mean?). If published, this will include your full peer review and any attached files.

Reviewer #1: No

Reviewer #2: No

Reviewer #3: Yes: Wojtek Przepiorka (Utrecht University, Department of Sociology / ICS)

Reviewer #4: No

---

## [Author Response · Author response to Decision Letter 0]

31 May 2020

Dear editor,

Thank you for letting us rework our article to improve it. We have changed a lot of things, as requested by the reviewers. It seems to us that the article is much clearer in this new version, with more justification and a better position compared to the existing literature.

Given these many changes, we could not attach the paper with the tracking changes, it was no longer readable, we created a new file. Consequently, the "Manuscript revised with tracking changes" file is the same as the "Manuscript" file.

We hope that this new version will meet the reviewers' expectations.

Sincerely yours

---

## [Decision Letter · Decision Letter 1]

23 Jun 2020

PONE-D-20-03051R1

Contrasting effects of information sharing on CPR extraction behaviour: experimental findings

PLOS ONE

Dear Dr. DUBOIS,

Thank you for submitting your manuscript to PLOS ONE. After careful consideration, we feel that it has merit but does not fully meet PLOS ONE’s publication criteria as it currently stands. Therefore, we invite you to submit a revised version of the manuscript that addresses the points raised during the review process.

We look forward to receiving your revised manuscript.

Kind regards,

Valerio Capraro

Academic Editor

PLOS ONE

Additional Editor Comments (if provided):

I have now collected three reviews from three of the four experts who reviewed the first version of this article (the fourth one declined my invitation). The reviewers think that the paper has improved, but two of them (and especially one of them) still think that more work should be done before the article can be published. Therefore, I invite you to revise your work again. Please follow the reviewers' comments closely.

Looking forward for the revision.

Reviewers' comments:

Reviewer's Responses to Questions

**Comments to the Author**

1. If the authors have adequately addressed your comments raised in a previous round of review and you feel that this manuscript is now acceptable for publication, you may indicate that here to bypass the “Comments to the Author” section, enter your conflict of interest statement in the “Confidential to Editor” section, and submit your "Accept" recommendation.

Reviewer #2: All comments have been addressed

Reviewer #3: (No Response)

Reviewer #4: All comments have been addressed

2. Is the manuscript technically sound, and do the data support the conclusions?

Reviewer #2: Yes

Reviewer #3: Partly

Reviewer #4: Yes

3. Has the statistical analysis been performed appropriately and rigorously? 

Reviewer #2: Yes

Reviewer #3: Yes

Reviewer #4: Yes

4. Have the authors made all data underlying the findings in their manuscript fully available?

Reviewer #2: Yes

Reviewer #3: Yes

Reviewer #4: Yes

5. Is the manuscript presented in an intelligible fashion and written in standard English?

Reviewer #2: Yes

Reviewer #3: Yes

Reviewer #4: Yes

6. Review Comments to the Author

Reviewer #2: The experiment is well-executed, the analysis well-performed and the results represent interesting contributions to behavioral and public economics, which are the main items for publication in PLoS ONE. The paper has significantly improved in the revision as a consequence of the comments of all the reviewers’ comments. Personally, I am fully satisfied with the answers of the authors. Maybe I would also state a minor comment before publication. Please, revise that there is a one-to-one correspondence between cites and references (I have detected that some cited papers are not included in the references, e.g. Lacomba et al., 2017).

Reviewer #3: The paper has much improved but it still requires much work. The authors did not sufficiently address all my points in their revision.

General comments

Information sharing as pro-social act: I am still not convinced that in the context of this experiment, sharing extraction information can be called pro-social in general. On page 4 the authors write that “sharing one's decision can also be seen as a pro-social act that enhances one's social image and self image”. Note, however, that the motive of enhancing one’s social image and/or self-image does not imply other-regarding preferences.

Lab experiments with economic games are not a standard approach in the social sciences an therefore the readership of PLOS ONE may not be entirely familiar with the terminology the authors use. Of course it is not necessary to explain everything from the start. I would only like to encourage the authors to read their paper through the lens of someone who is not an experimental economist and see whether everything is so self-explanatory (it is not).

The paper would benefit from having it edited by a professional English editing service.

If I remember correctly, PLOS ONE does not allow footnotes. The authors should check this and integrate the text they currently have in the footnotes in the main text.

Throughout the paper, subscript notation does not seem to be working out very well. The authors should take greater care when editing in-text formulas, equations, variables, etc.

There are a few points regarding the econometric model that are difficult to understand (see my particular comments on that).

Particular comments

Table 2 should mention in MD at Step 2 that players had no choice. Otherwise one might wonder what happens in this condition. Also in the text the authors should say that nothing happens in Step 2 of MD, and in Step 3 subjects get the round summary. It will be clearer and more consistent in this way.

It is still unclear how subjects were randomly assigned to experimental conditions (randomization, not random selection, is the quintessential precondition for causal inference in experiments). If randomization was not possible, the authors should mention it explicitly.

On page 12, the sentence “Experimental economics is very useful in this case too”, does not seem to be very useful.

Despite of what the authors write in their reply, the use of terminology is still not consistent. For example, in Figure 1, y-axes are labeled with “disclosure” and “sharing” although, in all likelihood, these terms are meant to say the same. Speaking of Figure 1 (and the other figures for that matter), the labels are very small, which makes it difficult to read the figures. The authors should try to improve this.

Bottom of page 13 and top of page 14, what does “statistic =” stand for? What kind of statistic do the authors report here (Chi2? F? t?). Were these tests comparing average extractions for all 20 rounds? And, please, do not use one-sided p-values. Use two-sided with alpha = 5% throughout the paper. This is standard (probably also in PLOS ONE).

Concerning the econometric model described on page 14, I was wondering why the authors have a lagged dependent as well as the total group extraction level from the previous round. At least the second variables should be total group extraction minus player i’s extraction. This will reduce collinearity and better allow to identify conditional cooperation.

What is the theoretical rationale for including decision time in the model. The authors need to justify that (I did not see they do that in the earlier parts of the paper).

On page 15/16, the authors write: “Finally, the estimation showed that, regardless of the treatment, the first decision in the game mattered. ” However, as far as I can see, this is not something they can conclude based on the estimation of the model shown in Table 4. To be able to say that there is a positive effect of the initial decision irrespective of treatment, there must be no significant interaction effects. But the authors do not estimate interactions between treatment variables and initial decisions. Moreover, they admit that these effects differ across treatment on page 19 (contradicting their initial statement).

Related to that, I was wondering how the authors, in the statistical model, distinguish between the individuals’ past decision in round 2 and these individuals’ initial decision. Maybe I missed in the text the authors’ explanations.

In figure 4, I find it interesting that the subjects that lie about their extractions lie increasingly more with the average increase of extraction levels. They seem to believe that once things go to “hell”, lying more about the true state of nature could maintain things in their favor. I am juts wondering whether the authors would like to say a bit more about this finding in the paper.

I did not find a discussion of the limitations of the authors approach in the conclusions section. From my previous review: “…the authors could discuss some obvious limitations of their study and point out directions for future research. As far as I can see, the main limitations of the study are: (1) there is no control condition without information provision, which would correspond to the standard nowadays (most people do not know how much their neighbors extract or what the average in their neighborhood or their village/ city is). (2) Even if free-riders did not disclose their extraction level, given the small groups (n =4) participants could infer the number of free riders from average or even own earnings in a round…”

Reviewer #4: The review work is mostly satisfying. However, I felt that limitations of the research is not yet fully explained such as the implementation of the mechanisms author propose in a dynamic environment shall be a very good idea for extension. However, for a first case study author have taken the static framework to make it easier and allows a better understanding of the behavioral dimension of the effects induced by these voluntary sharing mechanisms, which is understandable. Therefore, I think, author should still go ahead and report this.

7. PLOS authors have the option to publish the peer review history of their article (what does this mean?). If published, this will include your full peer review and any attached files.

Reviewer #2: No

Reviewer #3: Yes: Wojtek Przepiorka (Utrecht University, Department of Sociology / ICS)

Reviewer #4: No

---

## [Author Response · Author response to Decision Letter 1]

4 Aug 2020

Dear Editor, 

Thank you for letting us review our article. The reviewers' comments and suggestions really helped us improve the article.

As answered to Reviewer 3, it was not possible to send the article to a professional English-language editing service between the two revisions because of summer holidays (the university's administrative department is on leave). If the article is accepted, we commit to do so as soon as possible.

---

## [Decision Letter · Decision Letter 2]

20 Aug 2020

PONE-D-20-03051R2

Contrasting effects of information sharing on CPR extraction behaviour: experimental findings

PLOS ONE

Dear Dr. DUBOIS,

Thank you for submitting your manuscript to PLOS ONE. After careful consideration, we feel that it has merit but does not fully meet PLOS ONE’s publication criteria as it currently stands. Therefore, we invite you to submit a revised version of the manuscript that addresses the points raised during the review process.

We look forward to receiving your revised manuscript.

Kind regards,

Valerio Capraro

Academic Editor

PLOS ONE

Additional Editor Comments (if provided):

One of the reviewers suggests one last minor change. Please address this comment at your earliest convenience. I am looking forward for the final version.

Reviewers' comments:

Reviewer's Responses to Questions

**Comments to the Author**

1. If the authors have adequately addressed your comments raised in a previous round of review and you feel that this manuscript is now acceptable for publication, you may indicate that here to bypass the “Comments to the Author” section, enter your conflict of interest statement in the “Confidential to Editor” section, and submit your "Accept" recommendation.

Reviewer #2: All comments have been addressed

Reviewer #3: All comments have been addressed

2. Is the manuscript technically sound, and do the data support the conclusions?

Reviewer #2: Yes

Reviewer #3: Yes

3. Has the statistical analysis been performed appropriately and rigorously? 

Reviewer #2: Yes

Reviewer #3: Yes

4. Have the authors made all data underlying the findings in their manuscript fully available?

Reviewer #2: Yes

Reviewer #3: Yes

5. Is the manuscript presented in an intelligible fashion and written in standard English?

Reviewer #2: Yes

Reviewer #3: Yes

6. Review Comments to the Author

Reviewer #2: I am satisfied with the answers on my previous comments and I consider that the current paper deserves publication in PLOS ONE.

Reviewer #3: Many thanks for addressing the remaining issues I had with the paper. I think the paper has much improved once more.

For better readability of the results section, please consider adding references to the estimates and the corresponding statistics in the regression tables that you mention in the text in the text as well. Otherwise, the reader will not be able to follow how you used the regression models to test for significant differences etc.

And, although I think the text is pretty good in terms of language, it could benefit from having it read once more by an editing service or native speaker.

7. PLOS authors have the option to publish the peer review history of their article (what does this mean?). If published, this will include your full peer review and any attached files.

Reviewer #2: No

Reviewer #3: **Yes: **Wojtek Przepiorka (Utrecht University, Department of Sociology)

---

## [Author Response · Author response to Decision Letter 2]

20 Sep 2020

We have made the final minor changes suggested by Reviewer 3 and, thanks to him, the content of the document has been further improved.

---

## [Editor Report · Decision Letter 3]

23 Sep 2020

Contrasting effects of information sharing on CPR extraction behaviour: experimental findings

PONE-D-20-03051R3

Dear Dr. DUBOIS,

We’re pleased to inform you that your manuscript has been judged scientifically suitable for publication and will be formally accepted for publication once it meets all outstanding technical requirements.

Kind regards,

Valerio Capraro

Academic Editor

PLOS ONE
---

## [Editor Report · Acceptance letter]

28 Sep 2020

PONE-D-20-03051R3 

Contrasting effects of information sharing on common-pool resource extraction behavior: experimental findings 

Dear Dr. Dubois:

I'm pleased to inform you that your manuscript has been deemed suitable for publication in PLOS ONE. Congratulations! Your manuscript is now with our production department. 

Kind regards, 

on behalf of

Dr. Valerio Capraro 

Academic Editor

PLOS ONE